# A practical evaluation of statistical methods for the analysis of patient reported outcomes in an observational pharmaceutical study

Lucy R. Williams[1,2], Andrea Marongiu[1], Filippos T. Filippidis[2], Marion Heinzkill[1], Anna R. van Troostenburg[1], Richard Haubrich[3], Heribert Ramroth[1]*

**1** Gilead Sciences, Inc., Foster City, California, United States of America, **2** School of Public Health, White City Campus, Imperial College London, London, United Kingdom, **3** School of Medicine, University of California San Diego, San Diego, California, United States of America

* Heribert.Ramroth@gilead.com

## Abstract

### Background

Patient-reported outcomes (PROs) provide a unique opportunity to tailor clinical care to patients' needs. Observational pharmaceutical industry analyses of PROs in the HIV field often utilise simplistic pairwise comparisons of pre-defined follow-up periods to baseline, making inappropriate missing data assumptions and yielding limited information on the nature of the change in PRO. Our aim was to evaluate different statistical approaches for PRO analyses.

### Methods

Paired difference tests, Friedman's ANOVAs (F-ANOVA), linear mixed models (LMMs) and weighted generalised estimating equations (wGEEs) were applied to the analysis of the Short Form 36 (SF-36) mental component score (MCS) and physical component score (PCS) from treatment-naïve patients in an observational cohort of people living with HIV. Changes in MCS and PCS were assessed to compare the benefits of each approach.

### Results

The paired difference test demonstrated statistically significant increases in MCS and PCS from baseline to every follow-up, assuming however, data were missing completely at random. Use of the F-ANOVA was limited due to unbalanced data, leading to non-responder bias. While controlling for covariates, the LMMs and wGEEs illustrated a statistically significant increase in MCS and PCS with a steep increase over the first few months, followed by a plateau.

**Data availability statement:** Gilead Sciences shares anonymised individual patient data upon request or as required by law or regulation with qualified external researchers based on submitted curriculum vitae and reflecting non conflict of interest. The request proposal must also include a statistician. Approval of such requests is at Gilead Science's discretion and is dependent on the nature of the request, the merit of the research proposed, the availability of the data, and the intended use of the data. Data requests should be sent to datasharing@gilead.com.

**Funding:** Lucy Williams (LW) was a prior employee of Gilead Sciences. Richard Haubrich (RH), Andrea Marongiu (AM), Anna van Troostenburg (AV), Marion Heinzkill (AH), and Heribert Ramroth (HR) are current or former employees and stock owners of Gilead Sciences. The funder provided support in the form of salaries for the following authors [LW, RH, AM, MH, AV, HR]. However, the funder did not have an additional role in the analysis, decision to publish, or preparation of the presented methodological manuscript. All authors had a role in the study design, data collection and analysis, decision to publish, or preparation of the manuscript. The specific roles of these authors are articulated in the 'author contributions' section.

**Competing interests:** I have read the journal's policy and the authors of this manuscript have the following competing interests: Lucy Williams was a prior employee of Gilead Sciences. Richard Haubrich, Andrea Marongiu, Marion Heinzkill, Anna van Troostenburg, and Heribert Ramroth are current or former employees and stock owners of Gilead Sciences.

## Conclusion

Relative to paired difference tests, multivariable regression approaches can better handle missing data, control for confounding factors, and provide information on the timing and magnitude of PRO changes. Regression methods therefore facilitate more informative conclusions in observational PRO analyses, and thus provide more detailed evaluations of treatment regimens from the patient's perspective.

## Introduction

The use of patient-reported outcomes (PROs) in clinical practice and research allows patients' perspectives to be integrated into the evaluation of their treatment [1]. These outcomes are defined as any report of the status of a patient's health condition that comes directly from the patient, without interpretation by a healthcare professional [2]. They assess features of health beyond survival, laboratory tests and biomarkers, and address concepts that are either impossible or difficult to directly observe [3], such as treatment satisfaction [4] and health-related quality of life (HRQoL) [5]. PROs therefore present an opportunity to understand treatments from the patients' perspective in a more multidimensional manner than is possible with clinical outcomes alone.

The value of PROs has been recognised in clinical development, with PRO data being increasingly collected to better understand patients' treatment experiences [6,7]. However, concerns have been raised over their analysis and reporting [8,9], particularly regarding the handling of high rates of missing data, the use of unsuitable methods of statistical analysis and lack of consistency of analytical approaches across studies [10,11].

Missing data and unbalanced data (where there is an unequal number of observations across variable categories) are common problems in longitudinal studies as patients frequently miss visits or are lost to follow-up [12]. The missingness is rarely missing completely at random (MCAR; where missing data does not systematically differ from observed data [13]), and is more frequently missing at random (MAR; where missingness depends on observed data but not on unobserved data) or missing not at random (MNAR; where missingness is related to the unobserved responses) [14,15]. However simplistic statistical methods such as paired difference tests are often utilised that assume MCAR missingness [16,17], potentially leading to biased conclusions. Often these tests are used to repeatedly compare patient reported outcomes at each follow-up interval to baseline (e.g., treatment initiation) without adjustment for multiple testing.

To our knowledge, all work evaluating methods for the analysis of PROs have focussed on randomised controlled trials (RCTs). The aim of this study was to evaluate different statistical approaches for the analysis of numerical PROs using a real-world observational dataset. Given that the issues surrounding the statistical analysis of PROs have not yet been addressed in the field of HIV, and the increasing importance of their use for HIV patients [3], the statistical methods were evaluated on a HIV-1 dataset.

## Materials and methods

### Application dataset

The statistical methods were applied to PRO data collected in a prospective observational cohort study, "TAFNES" [GS-DE-292–1912], conducted between January 2016 and November 2019 [18]. Ethical clearance was given by the Ethical Committee of the Baden-Wuerttemberg State Medical Association, file number F-2015–075. Written informed consent was gained from patients included in the study. TAFNES enrolled 767 HIV-1 patients treated with Emtricitabine/Tenofovir alafenamide (F/TAF)-based antiretroviral treatments. PRO data were collected during routine visits at approximately 0, 3, 6, 12, 18 and 24 months after treatment initiation. As previous studies have shown greater change in PROs in treatment-naïve than treatment-experienced people living with HIV (PLWH) [18–20], the treatment-naive subgroup was selected for these analyses.

### Patient reported outcomes

Two PROs were evaluated: mental and physical HRQoL, collected using the Short Form-36 (SF-36) version 1 [21]. The SF-36 assesses HRQoL on eight scales: Physical Functioning, Role Physical, Bodily Pain, General Health, Vitality, Social Functioning, Role Emotional and Mental Health. These scales are used to compute two summary scores; the Mental Component Score (MCS) and the Physical Component Score (PCS), representing overall mental and physical HRQoL respectively. Each score ranges 0–100, with higher scores indicating higher HRQoL and a score of 50 representing the mean in the US calibration population.

### Statistical approaches

We evaluated four statistical approaches on the adequacy of their assumptions, the detail of their conclusions, and the communicability of their results (S1 Fig). As the true underlying change in MCS and PCS is unknown, we compare methods on their consistency with clinically plausible trends. The selected approaches were chosen because of the commonality of their use in HIV pharmaceutical PRO analyses [22–26], their applicability to skewed distributions that are characteristic of the SF-36 data [27,28], and their simplicity or ease of implementation.

1. Paired difference test (PD-test)

2. Friedman's ANOVA (F-ANOVA)

3. Linear mixed model (LMM)

4. Weighted generalised estimating equation (wGEE)

We selected methods that are either commonly used (PD-test, F-ANOVA) or have been proposed for PRO analyses but not frequently taken up (LMM, GEE) [1,26,39]. The wGEE analysis was extended to evaluate the use of a discrete- or continuous-time variable. We replicated the discrete-time wGEE model, using a continuous-time variable, and evaluated the following methods for non-linear trends:

1. Polynomial transformation

2. Fractional polynomial transformation

3. Piecewise linear splines

### Paired difference test

Paired difference tests compare the means or medians of two related samples. They are widely used in PRO analyses [22,23,29–31] due to their simplicity and the interpretability of their results, but they assume that missing data is MCAR.

We applied paired Wilcoxon rank sum (PWRS) tests because no assumptions are required on the normality of the out-come distribution. We tested validity of the MCAR assumption with Little's test [32] and then the significance of the differences between the MCS and PCS values, respectively, at M0 with their values at each visit window.

### Friedman's ANOVA

The non-parametric F-ANOVA tests whether two or more related population means are equal. It requires balanced data (the same number of observations at each factor level) and assumes that missing data are MCAR. The approach is occa-sionally used for PRO analyses [33,34], as it can compare PRO values at multiple time-points, whereas each paired difference test can be used on only two time-points. We applied the F-ANOVA to test the significance of the difference in PRO scores at each visit window. Where a significant difference was identified, post-hoc PWRS tests were used to identify the combinations of visit windows that were significantly different.

### Linear mixed model

The LMM is an extension of linear regression that tests the significance of the association between two variables. This maximum likelihood approach accounts for the correlation between repeated observations through the specification of random effects, and it can handle MCAR and MAR missingness. The approach is being increasingly used for the analysis of PROs in clinical trials, with a discrete-time variable [35].

To satisfy the LMM normality assumptions with the TAFNES SF-36 outcomes, both scores were log transformed: MCSt = (–ln(100 – MCS)) and PCSt = (–ln(100 – PCS). Both were modelled as a function of discrete-time, with the following baseline covariates: age, sex, HIV RNA count (log copies/ml), number of neuropsychiatric comorbidities, number of physical comorbidities, and presentation with advanced HIV, defined as persons presenting with a CD4 cell count ≤200/ul or presenting with an AIDS-defining event, regardless of the CD4 cell count [36]. The final combination of variables was selected using backwards selection (threshold p value 0.05) from a global model with all variables and their interactions with time. As backwards selection can lead to inappropriate removal of important covariates, we then validated the covariate selection using the Akaike information criterion (AIC) and including only those that improved the model fit (lower AIC). Age and sex were defined as a-priori covariates and were kept in the models regardless of statistical significance. A random effect for each individual was added to the intercept to account for correlation between repeated observations. Participants with missing covariate data were excluded from the LMM analysis.

### Weighted generalised estimating equation: discrete-time variable

The GEE is also an extension of the traditional linear model. It is a quasi-likelihood approach that accounts for correlation between observations with the specification of a working correlation matrix [37,38]. Although proposed as an appropriate method for PRO analyses [26], this approach has received little attention in practice [39].

As the traditional GEE assumes missing data are MCAR but the weighted GEE accounts for both MCAR and MAR missing data, we applied the wGEE. To weight observations by their inverse probability of being observed, weights were generated using multivariable logistic regression, as described in Salazar et al. [40]. A separate logistic regression model was executed for each visit window, using a binary outcome variable for whether an individual provided an SF-36 observation during a given follow-up interval or not. The predictors were the covariates detailed in section 3.2.3 as well as the most recently observed MCS and PCS values, where available. The fit of the models was checked with Hosmer-Lemeshow tests. Selection of the working correlation matrix was based on the quasi-likelihood under the independence model criterion (QIC), a measure of model fit with lower values representing better fit. The QIC was developed as a modification of the AIC to apply to models fit by the GEE approach. Therefore, in this manuscript we use the QIC for GEE analyses and AIC for LMM analyses. Participants with missing covariate data were excluded from the wGEE analysis.

### Weighted generalised estimating equation: continuous-time variable

To compare alternative non-linear modelling approaches for PRO analyses, the wGEE analysis was repeated with time as a continuous, numeric, non-linear variable. Models for MCS and PCS were fitted, modelling the time variable with polynomial [41], fractional polynomial [42] and piecewise linear spline [43] approaches. For each, the best-fitting functional form was first determined using the QIC in the full model and was then validated following covariate selection.

### Sensitivity analyses

While other GEE extensions are available, we focused on the weighted GEE as a relatively simple approach for handling MAR data. However, we performed a sensitivity analysis to compare the results of an unweighted GEE (with missing MCS and PCS values filled using multiple imputation) to the wGEE (S1 Methods).

Secondly, as previously recommended to reduce the number and strength of assumptions of more complex models, and to maximise interpretability [1], our primary analysis focussed on methods that assume MCAR or MAR missingness. Sensitivity analyses using multiple imputation was performed to evaluate the robustness of the LMM and wGEE results to MNAR data (S1 Methods).

### Software

All analyses were performed using R version 3.5.2. Sample R code is available in an online repository: https://github.com/lucyrose96/PRO-Methods-Sample-Code.

## Results

### Descriptive analysis

The sample consisted of 293 treatment-naïve participants. Median PCS at baseline was comparable to the general population (54.3, IQR 48.0–57.4), whereas MCS was lower (46.6, IQR 35.7–54.1). Each statistical approach's analysis population size and baseline characteristics differed to the overall recruited population to varying extents, depending on the missing data and balanced data assumptions (Table 1). Both MCS and PCS were negatively skewed over time (Shapiro-Wilk test P value <0.0001, S2 Fig).

While 269 patients (91.8%) provided evaluable SF-36 data at M0, this dropped to 185 (63.1%) at M3 and 163 (60.6%) at M24 (Fig 1A). The overall population MCS and PCS increased between M0 and M6, then displayed a plateau or slight decline (Fig 1 C-1D). By M24, the median MCS increased by 5.97 (12.8%), while the median PCS increased by 1.40 (2.6%).

### PWRS test

The PWRS test demonstrated a statistically significant increase in MCS and PCS between treatment initiation and every follow-up visit window (Table 2). As most participants had provided SF-36 data at baseline, the requirement of the PWRS test for balanced data reduced its analysis population size only slightly. However, there were differences between the populations included in the analyses. In the M0-M24 PWRS analysis population, the median baseline MCS was higher than in the population excluded and there were moderate differences in the presence of physical comorbidities. A significant result for Little's test (p=0.0001) showed that its MCAR assumption was not reasonable.

### Friedman's ANOVA

The requirement for balanced data (for our analysis, an observation at each visit window) in the F-ANOVA reduced its analysis population to 73. This reduced sample demonstrated differing trends to the excluded unbalanced population, leading the F-ANOVA tests to demonstrate contrasting results to the PWRS (S3 Fig). Although the F-ANOVA

 

**Table 1. Baseline demographic, clinical and SF-36 characteristics of the total analysis population and each statistical analysis population.**

| | Total Population [n = 293] | PWRS Test Population [n = 155(52.9%)] | | F-ANOVA Population [n = 73 (24.9%)] | | LMM and wGEE Populations [n = 285(97.3%)] | |
|---|---|---|---|---|---|---|---|
| | Value | Value | P value | Value | P value | Value | P value |
| Sex (Male) | 275 (93.9) | 150 (96.8) | 0.05 | 70 (95.9) | 0.58 | 267 (93.7) | 1.0 |
| Mean Age (SD) | 38.6 (11.0) | 38.8 (11.2) | 0.73 | 39.3 (10.6) | 0.51 | 38.6 (11.1) | 0.96 |
| Median HIV RNA, log c/ml (IQR) | 10.3 (9.2-11.9) | 10.3 (8.9-11.7) | 0.78 | 10.1 (8.9-11.7) | 0.53 | 10.3 (9.1-11.7) | 0.22 |
| <50 c/ml | 3 (1.0) | 1 (0.6) | – | 1 (1.4) | – | 3 (1.1) | – |
| >100,000 c/ml | 89 (30.4) | 51 (28.8) | – | 21 (28.8) | – | 83 (29.1) | – |
| Neuropsychiatric comorbidities | 0.2 (0.4) | 0.19 (0.43) | 0.66 | 0.19 (0.43) | 0.63 | 0.17 (0.42) | 0.72 |
| 0 | 248 (84.6) | 129 (83.2) | – | 60 (82.2) | – | 241 (84.6) | – |
| 1 | 40 (13.7) | 24 (15.5) | – | 12 (16.4) | – | 39 (13.7) | – |
| 2+ | 5 (1.7) | 2 (1.3) | – | 1 (1.4) | – | 5 (1.8) | – |
| Physical comorbidities [a] | 0.9 (1.3) | 1.07 (1.60) | 0.06 | 1.06 (1.60) | 0.17 | 0.87 (1.27) | 0.37 |
| 0 | 157 (53.6) | 73 (47.1) | – | 34 (46.6) | – | 151 (53.0) | – |
| 1 | 77 (26.3) | 47 (30.3) | – | 23 (31.5) | – | 76 (26.7) | – |
| 2+ | 59 (20.1) | 35 (22.6) | – | 16 (21.9) | – | 58 (20.4) | – |
| Advanced HIV disease (Yes) | 55 (18.8) | 27 (17.4) | 0.63 | 13 (17.8) | 0.94 | 54 (18.9) | 1.00 |
| Median MCS | 46.6 (35.7-54.1) | 47.4 (39.5-54.1) | 0.03 | 50.3 (40.4-55.4) | 0.03 | 46.6 (35.8-54.1) | NA |
| Median PCS | 54.3 (48.0-57.4) | 54.5 (48.0-57.8) | 0.46 | 54.1 (47.7-57.5) | 0.84 | 54.4 (48.0-57.4) | NA |

Value represents Number (%), unless otherwise stated. P value represents statistical significance of the difference between the analysis population and the non-analysis population. Continuous variables analysed with the student's t test, categorical variables with the chi squared test. PWRS = Paired Wilcoxon Rank Sum (Month 0 to Month 24 test), F-ANOVA = Friedman's Analysis of Variance, LMM = Linear Mixed Model, wGEE = weighted generalised estimating equation, SD = standard deviation, IQR = interquartile range, c/mL = copies per millilitre.

demonstrated a statistically significant difference in the median MCS between study visits (df = 5, chi = 15, p = 0.01), post-hoc tests did not find statistically significant differences between baseline and every follow-up. For PCS, the F-ANOVA did not identify a statistically significant difference in median scores between study visits (df = 5, chi = 6, p = 0.3).

## LMM and wGEE

The regression outputs for the LMM and wGEE demonstrated a significant increase in mental HRQoL from treatment initiation to every follow-up period, as well as an association with the number of ongoing neuropsychiatric comorbidities and log HIV RNA count at baseline (S1 Table). In both models, a higher number of neuropsychiatric comorbidities and higher log HIV RNA count at baseline were negatively associated with mental HRQoL, but the negative effect of initial HIV RNA count on MCS did not persist over the first few months of follow-up.

The PCS LMM and wGEE models showed a significant increase in PCS from treatment initiation to every follow-up period (S2 Table). Variables that were predictive of higher PCS were: age, number of ongoing physical comorbidities, presentation with advanced HIV and log HIV RNA at baseline. Although presentation with advanced HIV and higher HIV RNA at baseline were significantly associated with lower PCS at treatment initiation, the effects were reduced by the first follow-up visit at M3. LMM model diagnostic plots are provided in S4 Fig and the logistic regression results for the wGEE weighting models are presented in S3 Table. The unstructured working correlation matrix was selected for all wGEE models (S4 Table).

## wGEE continuous-time analysis

The best-fitting continuous-time MCS wGEE was the fractional polynomial $(((time+0.1)/100)^{-2}$, QIC = 5236), followed by the piece-wise linear spline (QIC = 5238), then the polynomial (QIC = 5239). For the PCS, the best-fitting model was the 3-degree polynomial

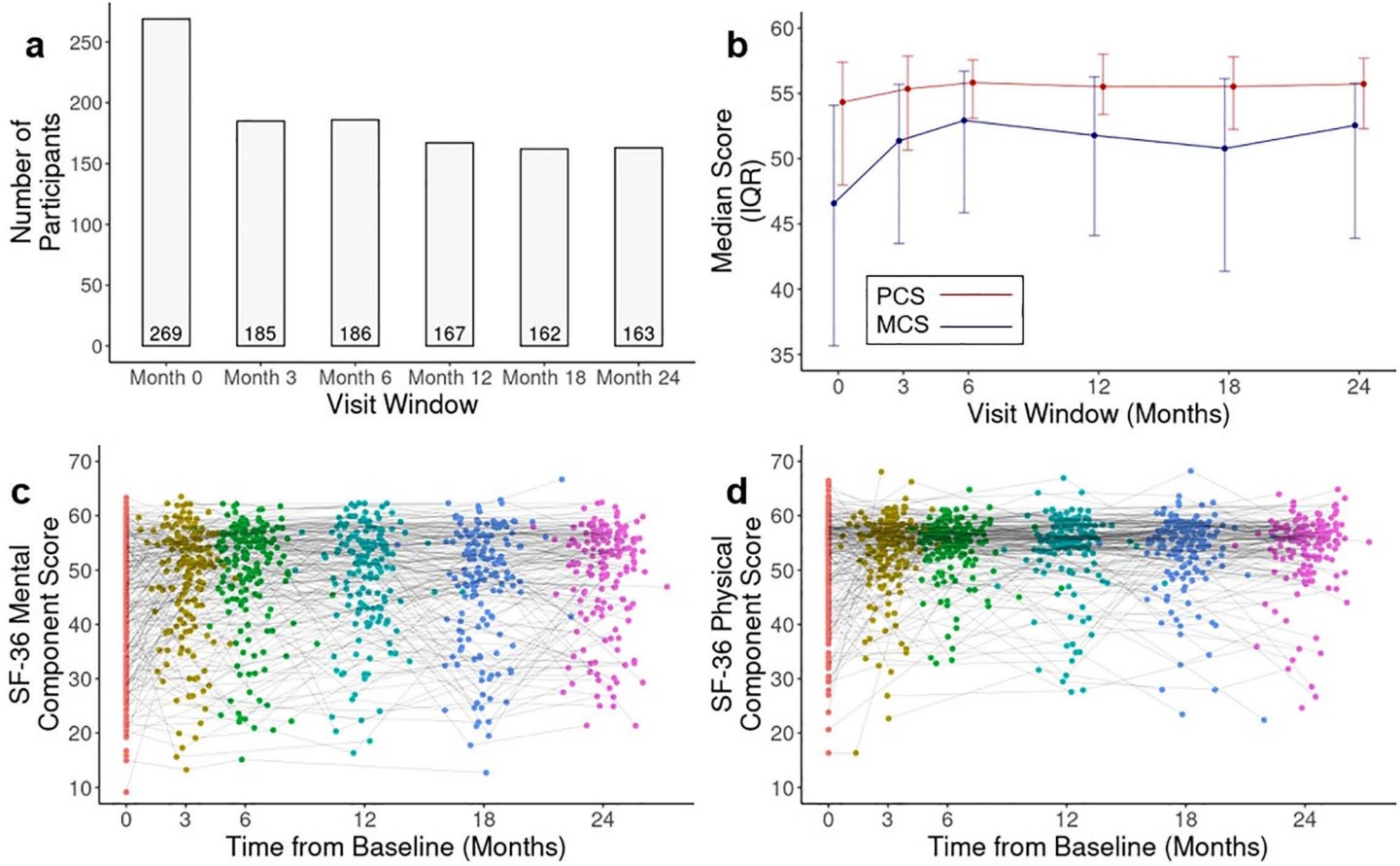

**Fig 1. Change in SF-36 mental component score (MCS) and physical component score (PCS) over the two-year follow-up. (A)** Number of individuals providing evaluable SF-36 data in each visit window. **(B)** Population-level change in median (Inter-Quartile Range) MCS (blue) and PCS (red). **(C)** Individual-level change in PCS. **(D)** Individual-level change in MCS. Points in **(C)** and **(D)** coloured by visit window assigned in data processing: Red=Baseline, Yellow=3 months, Green=6 months, Turquoise=12 months, Blue=18 months, Pink=24 months.

**Table 2. Paired Wilcoxon Rank Sum Test change in median mental and physical component scores.**

| Month 0 vs: | Analysis population size | Change in median PRO Score (P-value)[a] | |
|---|---|---|---|
| | | MCS | PCS |
| Month 3 | 177 | +3.9 (<0.001) | +1.0 (0.010) |
| Month 6 | 178 | +5.1 (<0.001) | +1.2 (0.006) |
| Month 12 | 158 | +4.6 (<0.001) | +0.6 (0.010) |
| Month 18 | 155 | +3.4 (<0.001) | +1.1 (0.010) |
| Month 24 | 155 | +5.1 (<0.001) | +1.3 (0.015) |

PRO=Patient Reported Outcome, MCS=Mental Component Score, PCS=Physical Component Score.

[a] Adjusted for multiple testing with the Benjamini-Hochberg Correction.

(QIC=4364), followed by the piecewise linear spline (QIC=4365), and the fractional polynomial (QIC=4367). All continuous MCS and PCS models gave a better fit than their respective categorical time models (MCS categorical QIC=5248, PCS categorical QIC=4374). However, the similarity of the QIC values demonstrate the similarity in the quality of model fit across approaches.

The best-fitting continuous-time models showed the same associations between the covariates and the outcomes that were previously identified in the categorical models (Tables 3 and 4). These models demonstrated the steep increase in the population-average mental HRQoL immediately after treatment initiation, as well as a more gradual incline in physical HRQoL, with a slight decline after the first year (Fig 2).

## Sensitivity analyses

The LMM and wGEE results were overall robust to missing data being MNAR. Analysis of the TAFNES data with an unweighted GEE with multiply imputed missing observations gave comparable results to the wGEE (S5 Table).

## Discussion

In this paper we evaluated four statistical approaches and within the last approach, three non-linear modelling approaches, for the analysis of longitudinal changes in PROs. Each statistical approach had generally consistent conclusions, but they differed in their depth, the appropriacy of their assumptions and the communicability of the results.

**Table 3. Weighted generalised estimating equation regression output for the Mental Component Score model with time modelled using a fractional polynomial.**

|  | Estimate | SE | Wald Statistic | P value |
|---|---|---|---|---|
| (Intercept) | 49.86 | 0.53 | 8749.25 | <0.001 |
| $(\text{Time [months]} + 0.1)/10)^{-2}$ | −4.27e-4 | 5.98e-5 | 56.80 | <0.001 |
| Sex (Female) | 0.41 | 1.91 | 0.05 | 0.832 |
| Age (decades) | 0.37 | 0.44 | 0.71 | 0.399 |
| Number of Neuropsychiatric Comorbidities | −6.87 | 1.37 | 25.31 | <0.001 |
| Log(HIV RNA) | −0.28 | 0.25 | 1.30 | 0.254 |
| $\text{x (Time} + 0.1)/10)^{-2}$ | −7.54e-5 | 2.62e-5 | 7.49 | 0.006 |

**Table 4. Weighted Generalised Estimating Equation regression output for the Physical Component Score model with time modelled with a three-degree polynomial.**

|  | Estimate | SE | Wald | P value |
|---|---|---|---|---|
| (Intercept) | 59.56 | 1.26 | 2240.39 | <0.001 |
| Time (months) | 0.43 | 0.15 | 8.09 | 0.004 |
| Time (months)$^2$ | −0.03 | 0.02 | 4.20 | 0.04 |
| Time (months)$^3$ | 6.60e-04 | 4.15e-04 | 2.54 | 0.111 |
| Sex (Female) | −4.68 | 2.59 | 3.27 | 0.071 |
| x Time (months) | 1.55 | 0.92 | 2.83 | 0.093 |
| x Time (months)$^2$ | −0.16 | 0.07 | 4.94 | 0.026 |
| x Time (months)$^3$ | 4.19e-03 | 1.49e-03 | 7.87 | 0.005 |
| Age (decades) | −1.59 | 0.33 | 23.20 | <0.001 |
| Number of Physical Comorbidities | −0.79 | 0.30 | 6.79 | 0.009 |
| Advanced HIV | −4.99 | 1.54 | 10.48 | 0.001 |
| x Time (months) | 1.48 | 0.45 | 10.78 | 0.001 |
| x Time (months)$^2$ | −0.11 | 0.04 | 6.29 | 0.012 |
| x Time (months)$^3$ | 2.07e-03 | 1.07e-03 | 3.73 | 0.053 |
| Log(HIV RNA) | −0.54 | 0.21 | 6.39 | 0.011 |
| x Time (months) | 0.21 | 0.06 | 11.40 | 0.001 |
| x Time (months)$^2$ | −0.02 | 0.01 | 8.94 | 0.003 |
| x Time (months)$^3$ | 4.51e-04 | 1.60e-04 | 7.99 | 0.005 |

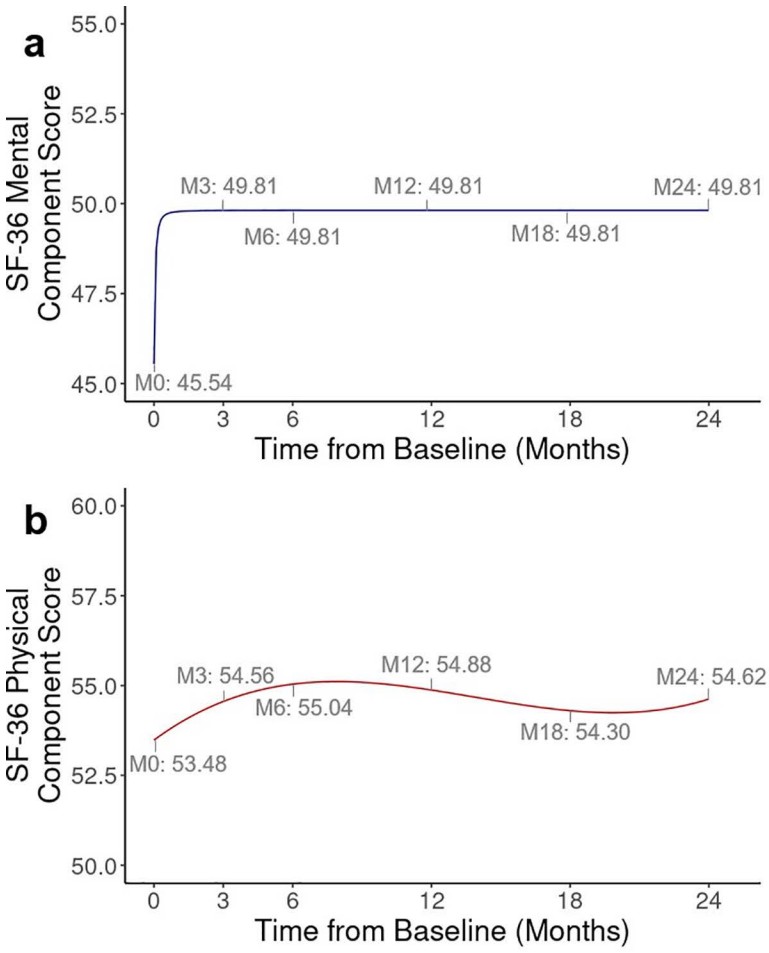

**Fig 2. Adjusted mental and physical component scores, estimated using the best-fitting non-linear weighted generalised estimating equation models. (A)** Mental component score (MCS). **(B)** Physical component score (PCS). MCS is modelled with time with a fractional polynomial [((time+0.1)/10)$^{-2}$], adjusting for sex, age and presentation with advanced HIV. PCS is modelled with a 3-degree polynomial [time3], adjusting for sex, age, physical comorbidities, presentation with advanced HIV and baseline HIV RNA count.

The strengths and limitations of each approach are summarised in Table 5. While the PD test is simple and easily interpreted, its missing data assumption was invalid, and it provided little insight into the longitudinal nature of the change in HRQoL and the factors that influenced it. The requirement of the F-ANOVA for balanced data limited its ability to accurately analyse the dataset, such that it gave contrasting results to all other approaches. This demonstrates a limitation of simpler statistical approaches; as they require data to be complete for all, or for certain combinations of visit windows, their sample size is reduced, and they are susceptible to non-responder bias. In contrast, the LMM and wGEE approaches utilised all SF-36 observations. They illustrated how mental and physical HRQoL changed after treatment initiation and identified factors associated with the changes. This is important for identifying clinical or socio-demographic groups with poorer HRQoL outcomes or poorer responses to treatment, and for controlling for variables associated with missingness. While it is possible to explore differences in changes in an outcome between subgroups with simpler statistical approaches such as the PD test, this requires stratification potentially to very small groups, leading to a large number of tests and increasing the likelihood of false positives. Multivariable regression approaches instead evaluate all covariates together. The LMM analyses required transformation of the MCS and PCS outcomes to meet its normality assumptions,

**Table 5. Summary of the strengths and limitations of the four statistical approaches.**

| Approach | | Strengths | Limitations |
|---|---|---|---|
| **Paired Wilcoxon Rank Sum test (PWRS-test)** | | • Easiest to perform and interpret | • Simplistic interpretation<br>• Inappropriate missing data assumption (MCAR)<br>• Compares only two visit windows at a time<br>• Requires balanced data, but only across two visit windows |
| **Friedman's ANOVA** | | • Compares all visit windows | • Inappropriate missing data assumption (MCAR)<br>• Requires balanced data across all visit windows, which results in reduction in sample size. |
| **Linear Mixed Model (LMM) and weighted Generalised Estimating Equation (wGEE)** | **Both** | • Can handle unbalanced data<br>• Large analysis population gives greater statistical power<br>• Additional covariate information<br>• Appropriate missing data assumption<br>• Can analyse continuous variables<br>• Can analyse interactions | • More complex to perform than PWRS test and Friedman's ANOVA |
| | **LMM only** | • Greater precision than wGEE | • Assumes normality of residuals and random effects ->outcome transformation required ->reduced interpretability |
| | **wGEE only** | • No distributional assumptions ->no outcome transformation required | • Lower precision than LMM<br>• Weighting required for appropriate missing data assumption |

and due to its complexity, the parameter estimates couldn't be back transformed so the interpretability of the model coefficients was reduced. This is a major limitation of LMMs when data do not satisfy its assumptions. In these cases, back-transforming model predictions and illustrating the change over time across patient profiles would support understanding. The issue with LMM interpretability contrasts to the more robust and flexible wGEE approach, which doesn't make distributional assumptions, therefore facilitating analysis on the original scale. As PRO data tend to be collected through questionnaires, they are susceptible to ceiling and floor effects (where most observed values are close to the maximum or minimum possible value, respectively). PRO data therefore often do not conform to the normal distribution, and in these cases, the wGEE may prove favourable.

The methods evaluated in this manuscript highlight an inherent trade-off in study design between model complexity/appropriateness of assumptions, and interpretability. Typically, simpler methods are more easily communicated, particularly to audiences without statistical training. More complex models may more accurately represent the data, but the underlying message can be lost. Ultimately, method selection should strike an appropriate balance for the study objectives.

A final benefit of both multivariable regression approaches was the ability to analyse continuous variables, where greater insight can be communicated on the nature of the change in a PRO over time. These analyses showed a steep incline in mental HRQoL soon after treatment initiation, but a more gradual change in physical HRQoL. As we used observational data, it is not possible to know if this trend represents the true underlying MCS and PCS trends, however it corroborates previously seen trends in physical and mental HRQoL following initiation of antiretrovirals in treatment-naïve PLWH [20]. Although the regression models themselves are more difficult to interpret, we recommend visualising trends through figures to aid communication of results to clinicians. From a statistical perspective, the continuous-time variable models had a better fit than the categorical-time models as they required fewer parameter estimates. This simplification is important for non-RCTs where confounders must be controlled for.

The fractional polynomial and polynomial models gave the best fitting models for the MCS and PCS respectively. However as both polynomial approaches use power terms to model non-linear trends, the parameter estimates themselves may be difficult to interpret for audiences without statistical training. In this illustration, we selected the best model based on model-fit,

however for other researchers, the more interpretable piecewise linear spline approach may be preferable, particularly if the model fit is similar to (fractional) polynomial transformations. If selecting a polynomial or fractional polynomial approach, PRO trends can be visualised using the estimates generated from the model to aid communication of results. This visualisation could also be extended to generate subgroup-specific estimates and confidence intervals could be generated with bootstrapping. The decision on the best approach for other datasets will depend on the underlying trend being modelled.

Our recommendation for the use of multivariable regression modelling approaches for the analysis of PROs is supported by recent work by the Setting International Standards for the Analysis of Quality of Life (SISAQOL) consortium. SISAQOL has discussed key statistical considerations for PRO analyses for cancer RCTs and highlight the importance of methods that can handle missing data appropriately, make realistic assumptions and produce interpretable results [1]. For evaluating the change in PRO at a time point and for describing its response trajectory over time, they recommend the use of an LMM with a discrete-time variable. While our work supports the use of multivariable regression approaches such as the LMM, we also highlight the value of more robust regression approaches such as the wGEE. Additionally, we demonstrate the improvement in model fit that can be made by modelling time continuously, particularly for observational studies where models may require multiple covariates and interactions, where the longitudinal change is of interest, and where patients might not attend follow-up visits at exact time points.

While the primary objective of this manuscript was to evaluate statistical approaches on data that reflect the complexity and variability of real-world clinical practice, simulations would be valuable to quantify the degree of accuracy and precision of each method. A simulation study was beyond the scope of this manuscript, but future work evaluating statistical methods on simulated PRO data would be valuable. The use of one example dataset may limit the generalisability of the conclusions of this research. PRO data comes in a variety of forms, so the benefits of one method for these data may not apply for other PRO data. However, the methods recommended in this paper are versatile for a range of outcomes; both the LMM and GEE can be applied to binomial (binary PROs, e.g., improvement/no improvement), Poisson (count PROs, e.g., number of symptoms), Gamma (exponentially distributed PROs, e.g., time to self-reported recovery) and gaussian data (numeric PROs, e.g., some score-based PROs). Importantly for PRO data, both can be applied to unbalanced data and data that are MAR or MCAR, and the GEE can be applied to skewed outcome distributions.

In this paper we have covered four statistical approaches, but depending on other researchers' data, other methods may be considered. For example, when data cannot be assumed to satisfy the MCAR or MAR assumptions, pattern-mixture models or joint modelling approaches can better handle missing data. Data that conforms to the normal distribution assumption with or without a link function can be handled with the LMM or generalised LMM, giving greater interpretability that when using the LMM with a complex transformation of the outcome variable. PROs may also be analysed as time to plateau or time to deterioration, in which case, survival-type approaches would be more appropriate than those covered in this manuscript. A wider range of methods have been discussed theoretically elsewhere [1], and future work would be valuable practically evaluating the applicability of these methods to PRO data.

## Conclusions

In this paper, we demonstrated the utility of multivariable regression modelling approaches for an example PRO dataset of HIV-1 patients, particularly for characterising the nature of the change in PRO over time, identifying associated covariates and for performing analysis with reasonable missing data assumptions. We highlight the benefit of the robustness of the wGEE for analysing non-normal data and recommend it as a favourable approach when the assumptions of the LMM cannot be met. Finally, we showed how modelling time continuously can improve the fit of covariate-heavy regression models.

## Supporting information

**S1 Methods. Supplementary Methods.**
(DOCX)

**S1 Fig. Analytical Processes.**
(DOCX)

**S2 Fig. Component Score Distributions.**
(DOCX)

**S3 Fig. Balanced and Unbalanced MCS and PCS Across Visits.**
(DOCX)

**S4 Fig. Model Diagnostics.**
(DOCX)

**S1 Table. Mental Component LMM and wGEE Regression Estimates.**
(DOCX)

**S2 Table. Physical Component LMM and wGEE Regression Estimates.**
(DOCX)

**S3 Table. Probability of SF-36 Observation Logistic Regression Results.**
(DOCX)

**S4 Table. Final wGEE QIC.**
(DOCX)

**S5 Table. Sensitivity Analyses.**
(DOCX)

## Acknowledgments

We thank Bo Zhao for conducting additional sensitivity analyses and Craig Pfeifer for helping with editorial aspects and submission of the manuscript.

## Author contributions

**Conceptualization:** Lucy R. Williams.

**Formal analysis:** Lucy R. Williams.

**Methodology:** Andrea Marongiu.

**Supervision:** Heribert Ramroth.

**Visualization:** Lucy R. Williams.

**Writing – original draft:** Lucy R. Williams, Andrea Marongiu, Heribert Ramroth.

**Writing – review & editing:** Lucy R. Williams, Andrea Marongiu, Filippos T. Filippidis, Marion Heinzkill, Anna R. van Troostenburg, Richard Haubrich, Heribert Ramroth.

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
