## [Decision Letter · Decision Letter 0]

7 Apr 2025

Dear Dr. Heribert Ramroth,

Thank you for submitting your manuscript to PLOS ONE. After careful consideration, we feel that it has merit but does not fully meet PLOS ONE’s publication criteria as it currently stands. Therefore, we invite you to submit a revised version of the manuscript that addresses the points raised during the review process.

We look forward to receiving your revised manuscript.

Kind regards,

Daniel Biftu Bekalo, PhD

Academic Editor

PLOS ONE

Journal Requirements:

2. Thank you for providing the following Funding Statement:

“Lucy Williams was a prior employee of Gilead Sciences.

Richard Haubrich, Andrea Marongiu, Marion Heinzkill, Anna van Troostenburg, and Heribert Ramroth are current or former employees and stock owners of Gilead Sciences.”

We note that one or more of the authors is affiliated with the funding organization, indicating the funder may have had some role in the design, data collection, analysis or preparation of your manuscript for publication; in other words, the funder played an indirect role through the participation of the co-authors.

If the funding organization did not play a role in the study design, data collection and analysis, decision to publish, or preparation of the manuscript and only provided financial support in the form of authors' salaries and/or research materials, please review your statements relating to the author contributions, and ensure you have specifically and accurately indicated the role(s) that these authors had in your study in the Author Contributions section of the online submission form. Please make any necessary amendments directly within this section of the online submission form.  Please also update your Funding Statement to include the following statement: “The funder provided support in the form of salaries for authors [insert relevant initials], but did not have any additional role in the study design, data collection and analysis, decision to publish, or preparation of the manuscript. The specific roles of these authors are articulated in the ‘author contributions’ section.”

If the funding organization did have an additional role, please state and explain that role within your Funding Statement.

Please also provide an updated Competing Interests Statement declaring this commercial affiliation along with any other relevant declarations relating to employment, consultancy, patents, products in development, or marketed products, etc.

Reviewers' comments:

Reviewer's Responses to Questions

**Comments to the Author**

1. Is the manuscript technically sound, and do the data support the conclusions?

Reviewer #1: Yes

Reviewer #2: Yes

2. Has the statistical analysis been performed appropriately and rigorously?

Reviewer #1: Yes

Reviewer #2: Yes

3. Have the authors made all data underlying the findings in their manuscript fully available?

Reviewer #1: No

Reviewer #2: No

4. Is the manuscript presented in an intelligible fashion and written in standard English?

Reviewer #1: Yes

Reviewer #2: Yes

Reviewer #1: The paper “A practical evaluation of statistical methods for the analysis of patient reported outcomes in an observational pharmaceutical study” is well-written, organized and statistically sound paper. It provides a comparative template of how longitudinal patient reported outcomes in HIV studies can be analyzed using four different statistical methods and which one should be chosen. However, the paper lacks novelty and reiterates what already exists in the literature. The authors have highlighted the assumptions and requirements of the methods, how these assumptions are violated or unmet in the PRO analyses, and which method can best handle them. But this is the very reason why the methods are being developed in the first place, and a study statistician analyzing the PRO data is expected to be aware of the assumptions and appropriateness of these methods in the study. Moreover, the pharmaceutical industry is well-aware of these methods and there appropriateness to their studies. Hence, this paper may be useful (if there is any) to a very limited audience working in the HIV studies with PROs.

Reviewer #2: General assessment

This manuscript addresses a highly relevant and timely topic in medical statistics and health outcomes research: the comparative evaluation of statistical methods for the analysis of patient-reported outcomes (PROs) in an observational HIV cohort. The authors present a practical application of four different methods (paired difference tests, Friedman’s ANOVA, linear mixed models – LMM, and weighted generalized estimating equations – wGEE) to assess longitudinal changes in SF-36 physical and mental health scores among treatment-naïve people living with HIV.

The paper is clearly written, and the methods are well motivated and generally appropriate for the stated aims. However, several important methodological and practical aspects of the statistical analysis require further clarification and strengthening, particularly in terms of assumptions, model diagnostics, missing data handling, and interpretability of results.

Major comments

1. Statistical assumptions and model validation

While the manuscript appropriately highlights the violation of the MCAR assumption (via Little’s test) for the paired difference tests, there is no presentation of diagnostic checks for key assumptions underlying the LMM and GEE models.

Recommendation: Please include diagnostic plots or summaries to assess the normality of residuals, adequacy of random effects distribution, and the correlation structure used in the GEE framework.

2. Transformations and interpretability

The log-transformation of the SF-36 outcomes for LMM (–ln(100 – score)) is statistically valid in the presence of ceiling effects but substantially reduces the interpretability of the resulting model coefficients.

Recommendation: Discuss this limitation more explicitly. Consider whether models such as GLMMs or quantile regression could provide more interpretable results without requiring transformation.

3. Model selection and covariate handling

Covariates were selected using backward elimination based on p-values and AIC. Although common in practice, this approach can be unstable and data-dependent, especially in observational settings with potential confounding.

Recommendation: Consider validating the selected models via cross-validation or information-theoretic criteria across multiple imputations. Additionally, clarify how missing covariate data were handled.

4. Weighted GEE models and missing data handling

The use of inverse probability weighting to adjust for missingness is a strength of the paper. However, the logistic models used to compute weights are not reported in detail (e.g., predictor coefficients, diagnostics, AUC, Hosmer-Lemeshow tests).

Recommendation: Please provide summary tables or appendices reporting these models and their diagnostics. The reproducibility and credibility of the wGEE results depends on the robustness of this step.

5. Model comparison and time modeling

The manuscript uses QIC to compare models with different time specifications. However, differences in QIC values are minimal, and the choice of the final model (e.g., fractional polynomial vs spline) is not strongly justified.

Recommendation: Discuss whether differences in model fit are statistically and practically meaningful. The use of simpler, more interpretable models (e.g., splines) may be preferable unless the polynomial forms provide substantial gains.

Minor comments

Figures and tables:

- Figure 2 could be enhanced with 95% confidence bands or bootstrapped intervals.

- Table 1 is informative but could include p-values comparing included vs excluded patients across methods to quantify potential bias due to differential inclusion.

Software and reproducibility:

The authors refer to a GitHub repository, which is commendable. However, more specific references to scripts corresponding to each analysis would facilitate replication.

Sensitivity to MNAR:

- The Authors briefly mention MNAR scenarios but only conduct sensitivity analyses under MAR assumptions. Approaches such as pattern-mixture models or joint modeling frameworks might provide deeper insight.

- Recommendation: If it is not feasible to implement, acknowledge this as a limitation and suggest directions for future work.

Conclusion

This manuscript provides a valuable and practical comparison of statistical methods for the analysis of PROs in observational settings. The use of real-world data, a comprehensive set of methods, and sensitivity analyses is commendable. However, the paper would greatly benefit from:

- enhanced reporting of model diagnostics and assumptions,

- justification of transformation choices and model selection,

- more detailed presentation of the weighting mechanism in wGEE.

After addressing these points, the manuscript will make a meaningful contribution to the applied statistical literature on longitudinal PRO analysis in HIV care and beyond.

References for Reviewer Comments

1. Fitzmaurice GM, Laird NM, Ware JH. Applied Longitudinal Analysis. 2nd ed. Wiley; 2012.

2. Verbeke G, Molenberghs G. Linear Mixed Models for Longitudinal Data. Springer; 2000.

3. Salazar A et al. Simple GEEs and wGEEs in longitudinal studies with dropouts. Statistics in Medicine. 2016;35:3424–3448.

4. Carpenter JR, Kenward MG. Multiple Imputation and Its Application. Wiley; 2013.

5. Royston P, Altman DG. Regression using fractional polynomials. Applied Statistics. 1994;43:429–467.

**Do you want your identity to be public for this peer review?** For information about this choice, including consent withdrawal, please see our Privacy Policy

Reviewer #1: No

Reviewer #2: No

---

## [Author Response · Author response to Decision Letter 1]

8 Sep 2025

I have uploaded the response to reviewers letter during file upload. I have also copied the text from that document here:

Dear PLOS One Editor and Reviewers,

Thank you for your review of the manuscript “A practical evaluation of statistical methods for the analysis of patient reported outcomes in an observational pharmaceutical study”. We appreciate the time you have taken on the review and the comments you provided.

Below we have provided responses to your comments.

Thank you for your time.

Best,

Heribert Ramroth

----

Reviewer #1

General assessment

The paper “A practical evaluation of statistical methods for the analysis of patient reported outcomes in an observational pharmaceutical study” is well-written, organized and statistically sound paper. It provides a comparative template of how longitudinal patient reported outcomes in HIV studies can be analyzed using four different statistical methods and which one should be chosen. However, the paper lacks novelty and reiterates what already exists in the literature. The authors have highlighted the assumptions and requirements of the methods, how these assumptions are violated or unmet in the PRO analyses, and which method can best handle them. But this is the very reason why the methods are being developed in the first place, and a study statistician analyzing the PRO data is expected to be aware of the assumptions and appropriateness of these methods in the study. Moreover, the pharmaceutical industry is well-aware of these methods and there appropriateness to their studies. Hence, this paper may be useful (if there is any) to a very limited audience working in the HIV studies with PROs.

Response: We thank reviewer one for the comment. We acknowledge that statisticians may be aware of the various methodologies, however, we disagree that this manuscript has a limited audience. The manuscript uses HIV data as an example of how to apply various methodologies for the analysis of PRO information. This does not limit the impact or applicability of the methodology to other therapeutic areas or to other industries (e.g., health care organizations, research institutions, etc.). The manuscript provides a methodologic comparison that will enable the wider audience to understand what direction would be appropriate for their analyses of PRO data.

Furthermore, the GEE approach is rarely used in the PRO field, despite its methodological advantages. We therefore would argue that the paper is novel and valuable in its illustration of the utility of the GEE, and that the code would be helpful for any statistician applying or adapting the method for a range of PRO data.

----

Reviewer #2:

General assessment

This manuscript addresses a highly relevant and timely topic in medical statistics and health outcomes research: the comparative evaluation of statistical methods for the analysis of patient-reported outcomes (PROs) in an observational HIV cohort. The authors present a practical application of four different methods (paired difference tests, Friedman’s ANOVA, linear mixed models – LMM, and weighted generalized estimating equations – wGEE) to assess longitudinal changes in SF-36 physical and mental health scores among treatment-naïve people living with HIV.

The paper is clearly written, and the methods are well motivated and generally appropriate for the stated aims. However, several important methodological and practical aspects of the statistical analysis require further clarification and strengthening, particularly in terms of assumptions, model diagnostics, missing data handling, and interpretability of results.

Response: Thank you for your comments. The commentary regarding further clarifications is presented below.

----

Major Comment 1. Statistical assumptions and model validation

While the manuscript appropriately highlights the violation of the MCAR assumption (via Little’s test) for the paired difference tests, there is no presentation of diagnostic checks for key assumptions underlying the LMM and GEE models.

Recommendation: Please include diagnostic plots or summaries to assess the normality of residuals, adequacy of random effects distribution, and the correlation structure used in the GEE framework.

Response: We thank reviewer two for their suggestion. We have added model diagnostic information of the linear mixed models to the Supplementary Figure S4, and referenced in the main text, line 231 “LMM model diagnostic plots are provided in Figure S4...”.

Please note that while there is some deviation at the tails of the distributions in the normality of residuals and random effects plots, the selected log transformation (PCSt = (–ln(100–PCS) and MCSt = (–ln(100–MCS)) was chosen because it satisfied the model assumptions most adequately from all transformations that were tested.

The only assumption of the generalised estimating equation (GEE) is that the working correlation matrix is correctly specified. We based our matrix selection on the model quasi-likelihood under the independence model criterion (QIC), with the lowest value representing the best model fit. This is a common approach for working correlation matrix selection [1]. As the unstructured matrix had the lowest QIC values, we selected this matrix for the MCS and PCS models. We have added a table of QIC values for each of the working correlation matrices to the Supplementary Materials: Supplementary Table 3, and referenced this in the main text, line 233: “The unstructured working correlation matrix was selected for all wGEE models (Table S4).”

----

Major comment 2. Transformations and interpretability

The log-transformation of the SF-36 outcomes for LMM (–ln(100 – score)) is statistically valid in the presence of ceiling effects but substantially reduces the interpretability of the resulting model coefficients.

Recommendation: Discuss this limitation more explicitly. Consider whether models such as GLMMs or quantile regression could provide more interpretable results without requiring transformation.

Response: We agree with Reviewer two that the log-transformation of the LMM is a major limitation of this statistical approach, as the model coefficients must be presented on this transformed scale. To improve the interpretability, a statistician may wish to generate back-transformed predictions for different patient profiles for the change in PRO over time (similar to the wGEE results presented in Figure 2). However, this may not be preferable if the model coefficients themselves are of interest, and in these cases, alternative approaches may be better, such as the (w)GEE, GLMMs or quantile regression.

We have added further elaboration on the limited interpretability of LMMs with a transformed outcome variable to the discussion, line 280: “The LMM analyses required transformation of the MCS and PCS outcomes to meet its normality assumptions, and due to its complexity, the parameter estimates couldn’t be back transformed so the interpretability of the model coefficients was reduced. This is a major limitation of LMMs when data do not satisfy its assumptions. In these cases, back-transforming model predictions and illustrating the change over time across patient profiles would support understanding”.

We have also noted where alternative methods to those illustrated in the manuscript to the discussion may be useful, line 331: “In this paper we have covered four statistical approaches, but depending on other researchers’ data, other methods may be considered. For example, when data cannot be assumed to satisfy the MCAR or MAR assumptions, pattern-mixture models or joint modelling approaches can better handle missing data. Data that conforms to the normal distribution assumption with or without a link function can be handled with the LMM or generalised LMM, giving greater interpretability that when using the LMM with a complex transformation of the outcome variable. A wider range of methods have been discussed theoretically elsewhere [1]. Future work would be valuable practically evaluating the applicability of these methods to PRO data.”

----

Major comment 3. Model selection and covariate handling

Covariates were selected using backward elimination based on p-values and AIC. Although common in practice, this approach can be unstable and data-dependent, especially in observational settings with potential confounding.

Recommendation: Consider validating the selected models via cross-validation or information-theoretic criteria across multiple imputations. Additionally, clarify how missing covariate data were handled.

Response: We appreciate the limitations of backwards selection and have now added a note on this to the methods, line 142: “As backwards selection can lead to inappropriate removal of important covariates, we then validated the covariate selection using the Akaike information criterion…”. Validating the models using cross-validation or multiple imputation would require more analysis time which we cannot allocate to this project. The AIC validation step only modified the covariate selection slightly, so we feel that the approach used is appropriate in this case.

Missing covariate data were handled using complete case analysis. 8 of 293 participants had missing covariate data, therefore we expect that this made little difference to the multivariable model results. We have noted this in the methods, lines 147 and 161: “Participants with missing covariate data were excluded from the LMM/wGEE analysis.”

----

Major Comment 4. Weighted GEE models and missing data handling

The use of inverse probability weighting to adjust for missingness is a strength of the paper. However, the logistic models used to compute weights are not reported in detail (e.g., predictor coefficients, diagnostics, AUC, Hosmer-Lemeshow tests).

Recommendation: Please provide summary tables or appendices reporting these models and their diagnostics. The reproducibility and credibility of the wGEE results depends on the robustness of this step.

Response: We thank reviewer two for their suggestion. We have added the model results and diagnostics to the Supplementary Table S3, and referenced this in the main text, line 231 “and the logistic regression results for the wGEE weighting models are presented in Table S3”. We have provided the results for the baseline (Month 0) model, which included the following predictor variables: HIV RNA (log10) at baseline, sex, age, number of neuropsychiatric comorbidities, number of physical comorbidities, and presentation with advanced HIV at baseline (yes/no). We have also provided the results for the Month 3, 6, 12, 18, and 24 models, which also included a variable for the most recently observed MCS and PCS score. Please note that for participants who did not have a value for the most recently observed MCS/PCS for a given follow-up visit window, we ran a separate model without these variables, and assigned the weight generated from this model for that participant’s visit. However, for simplicity, we have only presented the model results for this with the most recent MCS/PCS predictors. For each model, we performed the:

Hosmer-Lemeshow test – p value was greater than 0.05 for all models, indicating no significant difference between observed and expected outcomes, and therefore that the logistic regression model fits the data well.

Link function test - The second-order linear predictor was not statistically significant for all models, suggesting that the link function used (logit link) adequately captured the relationship between the predictors and the outcome.

We are therefore confident that the wGEE results are credible and reproducible.

----

Major comment 5. Model comparison and time modeling

The manuscript uses QIC to compare models with different time specifications. However, differences in QIC values are minimal, and the choice of the final model (e.g., fractional polynomial vs spline) is not strongly justified.

Recommendation: Discuss whether differences in model fit are statistically and practically meaningful. The use of simpler, more interpretable models (e.g., splines) may be preferable unless the polynomial forms provide substantial gains.

Response: We thank reviewer two for their comment. We recognise that there is little difference in QIC values across non-linear modelling approaches. We opted for the best-fitting model as this gave us a quantitative method of selection, compared basing our selection on interpretability, which would have been subjective. We agree that when other researchers are selecting models for their data, interpretability will be a priority and therefore simpler models may be more useful.

We have added some elaboration to our paragraph discussing this point in the discussion, line 300:

“The fractional polynomial and polynomial models gave the best fitting models for the MCS and PCS respectively. However, as both polynomial approaches use power terms to model non-linear trends, the parameter estimates themselves may be difficult to interpret for audiences without statistical training. In this illustration, we selected the best model based on model-fit, however for other researchers, the more interpretable piecewise linear spline approach may be preferable, particularly if the model fit is similar to (fractional) polynomial transformations. If selecting a polynomial or fractional polynomial approach, PRO trends can be visualised using the estimates generated from the model to aid communication of results. This visualisation could also be extended to generate subgroup-specific estimates and confidence intervals could be generated with bootstrapping. The decision on the best approach for other datasets will depend on the underlying trend being modelled.”

----

Minor comment 1. Figures and tables:

Figure 2 could be enhanced with 95% confidence bands or bootstrapped intervals.

Table 1 is informative but could include p-values comparing included vs excluded patients across methods to quantify potential bias due to differential inclusion.

Response: We thank the reviewer for their suggestion to improve figure 2, and we agree that 95% confidence intervals produced by bootstrapping would enhance the figure. However, due to time and resource constraints we are not able to generate these. We believe that the main messages of the figures are still communicated adequately, and we note that other statisticians applying these methods may choose to generate confidence intervals on predictions (line 307): “This visualisation could also be extended to generate subgroup-specific estimates and confidence intervals could be generated with bootstrapping”. We have added p-values to Table 1.

----

Minor comment 2. Software and reproducibility:

The authors refer to a GitHub repository, which is commendable. However, more specific references to scripts corresponding to each analysis would facilitate replication.

Response: The GitHub repository has each of the analyses clearly detailed in well-annotated, separate R markdown files (https://github.com/lucyrose96/PRO-Methods-Sample-Code). We therefore believe that it is sufficient to simply link to the overall repository and allow the reader to select the analysis of interest.

----

Minor comment 3. Sensitivity to MNAR:

- The Authors briefly mention MNAR scenarios but only conduct sensitivity analyses under MAR assumptions. Approaches such as pattern-mixture models or joint modeling frameworks might provide deeper insight.

- Recommendation: If it is not feasible to implement, acknowledge this as a limitation and suggest directions for future work.

Response: As suggested by Reviewer 2, implementing these models would be time and resource intensive, and we believe beyond the scope of this manuscript. We have instead added a comment to the discussion, line 330, recognising that under scenarios where the missing not at random (MNAR) assumption is valid, other models would be more appropriate.

“In this paper we have covered four statistical approaches, but depend

---

## [Decision Letter · Decision Letter 1]

1 Dec 2025

Thank you for submitting your manuscript to PLOS ONE. After careful consideration, we feel that it has merit but does not fully meet PLOS ONE’s publication criteria as it currently stands. Therefore, we invite you to submit a revised version of the manuscript that addresses the points raised during the review process.

We look forward to receiving your revised manuscript.

Kind regards,

Daniel Biftu Bekalo, PhD

Academic Editor

PLOS ONE

Journal Requirements:

Reviewers' comments:

Reviewer's Responses to Questions

**Comments to the Author**

Reviewer #1: All comments have been addressed

Reviewer #3: (No Response)

2. Is the manuscript technically sound, and do the data support the conclusions?

Reviewer #1: Yes

Reviewer #3: Yes

3. Has the statistical analysis been performed appropriately and rigorously?

Reviewer #1: Yes

Reviewer #3: Yes

4. Have the authors made all data underlying the findings in their manuscript fully available?

Reviewer #1: Yes

Reviewer #3: No

5. Is the manuscript presented in an intelligible fashion and written in standard English?

Reviewer #1: (No Response)

Reviewer #3: Yes

Reviewer #1: (No Response)

Reviewer #3: (No Response)

**Do you want your identity to be public for this peer review?** For information about this choice, including consent withdrawal, please see our Privacy Policy

Reviewer #1: No

Reviewer #3: No

---

## [Author Response · Author response to Decision Letter 2]

23 Dec 2025

Response to reviewers has been uploaded as part of the submission. Due to formatting issues within text boxes, I recommend reviewing the letter attachment. However, I have copied the contents of the letter here.

Dear PLOS One Editor and Reviewers,

Thank you for your review of the manuscript “A practical evaluation of statistical methods for the analysis of patient reported outcomes in an observational pharmaceutical study”. We appreciate the time you have taken on the review and the comments you provided.

Below we have provided responses to your comments.

Thank you for your time.

Best,

Heribert Ramroth

Reviewer’s comments:

Thank you for the opportunity to review this interesting and well-prepared manuscript. The study provides a clear and structured comparison of statistical methods commonly used for analyzing longitudinal patient-reported outcomes (PROs) in an observational cohort of HIV-infected patients. The manuscript is technically sound, the methods are rigorously implemented, and the results support the conclusions drawn. The writing is clear and of high quality.

Below, I provide comments aimed at enhancing the conceptual clarity and interpretive transparency of the manuscript.

Key Comments

Clarify the implicit baseline model used to evaluate the performance of statistical methods.

Throughout the manuscript, method comparisons are partly interpreted in relation to an expected clinical trajectory (e.g., an initial improvement in MCS/PCS followed by a plateau). While this expectation is consistent with previous literature, it effectively serves as an implicit baseline for assessing method appropriateness. Because the true underlying trajectory is unknown, it would be helpful to explicitly acknowledge that method performance is evaluated in terms of consistency with clinically plausible trends, rather than accuracy relative to a known ground truth.

Response: We thank the reviewer for their helpful observation. We agree that our comparisons implicitly relied on clinically plausible expectations of change (e.g., initial improvement in MCS/PCS followed by a plateau), as suggested by previous literature. To clarify this, we have revised the manuscript to explicitly state that method performance was evaluated in terms of consistency with these expected trends rather than accuracy against a known ground truth, since the true underlying trajectory is unknown in observational HIV studies. The revised text is in the Methods, line 103: “As the true underlying change in MCS and PCS is unknown, we compare methods on their consistency with clinically plausible trends”, and the Discussion, line 308: “As we used observational data, it is not possible to know if this trend represents the true underlying MCS and PCS trends, however it corroborates previously seen trends in physical and mental HRQoL following initiation of antiretrovirals in treatment-naïve PLWH”.

Lack of an objective performance benchmark.

Because the analysis relies exclusively on real observational data, it is inherently descriptive. While appropriate for the stated objectives, the lack of simulation-based analyses limits the ability to quantify the bias or robustness of each method in controlled missingness scenarios. Even a brief simulation (e.g., in the Supplementary Material) using known missingness trajectories and mechanisms would substantially strengthen the methodological conclusions.

Response: We agree with the reviewer that in order to quantify the degree of bias and accuracy of each statistical approach, a simulation study would be desirableis required. To do this properly, a range of plausible scenarios should be simulated to test the sensitivity of the results to the scenario that is given. It would require significant time and resource to test the alternative MCS/PCS trajectories, intermittent and dropout missingness, skewed data distributions, etc. We believe that this would also be a valuable contribution to the literature but is ultimately enough content for a separate manuscript. For our manuscript, the primary objective was to evaluate statistical approaches on data that reflect the complexity and variability of real-world clinical practice.

We have therefore instead added some text to the Discussion, line 337, to recommend simulations in future studies: “While the primary objective of this manuscript was to evaluate statistical approaches on data that reflect the complexity and variability of real-world clinical practice, simulations would be valuable to quantify the degree of accuracy and precision of each method. A simulation study was beyond the scope of this manuscript, but future work evaluating statistical methods on simulated PRO data would be valuable.”

Contextualization of the selected methods.

The manuscript focuses on four approaches commonly used in practice, which is reasonable. However, a brief explanation of why other families of longitudinal models (e.g., nonlinear mixed models, joint models, or survival-type "time-to-plateau" approaches) were not considered would help define the scope of the work and guide future extensions.

Response: We selected methods that are either commonly used (paired difference test, ANOVA) or have been proposed for PRO analyses but not frequently taken up (LMM, GEE). We have added this contextualisation when introducing the selected approaches in the Methods, line 112: “We selected methods that are either commonly used (PD-test, F-ANOVA) or have been proposed for PRO analyses but not frequently taken up (LMM, GEE) [1,26,39].”. We also previously had a paragraph discussing alternative approaches that may be favourable for other datasets. We have elaborated on this paragraph with the methods suggested by this reviewer. Discussion, line 348:

“In this paper we have covered four statistical approaches, but depending on other researchers’ data, other methods may be considered. For example, when data cannot be assumed to satisfy the MCAR or MAR assumptions, pattern-mixture models or joint modelling approaches can better handle missing data. Data that conforms to the normal distribution assumption with or without a link function can be handled with the LMM or generalised LMM, giving greater interpretability that when using the LMM with a complex transformation of the outcome variable. PROs may also be analysed as time to plateau or time to deterioration, in which case, survival-type approaches would be more appropriate than those covered in this manuscript. A wider range of methods have been discussed theoretically elsewhere [1], and future work would be valuable practically evaluating the applicability of these methods to PRO data.”

Interpretation of LMM Transformations.

The need to log-transform the results for LMM normality assumptions reduces interpretability. While this limitation is acknowledged, I suggest highlighting it more explicitly as part of the trade-off between model assumptions and interpretability, especially since one of the objectives of the manuscript is to assess the communicability of the results.

Response: We thank the reviewer for their suggestion. We would like to highlight that we have addressed this in detail in the discussion, line 287:

“The LMM analyses required transformation of the MCS and PCS outcomes to meet its normality assumptions, and due to its complexity, the parameter estimates couldn’t be back transformed so the interpretability of the model coefficients was reduced. This is a major limitation of LMMs when data do not satisfy its assumptions. In these cases, back-transforming model predictions and illustrating the change over time across patient profiles would support understanding. The issue with LMM interpretability contrasts to the more robust and flexible wGEE approach, which doesn’t make distributional assumptions, therefore facilitating analysis on the original scale”

However, we appreciate the suggestion to frame it within the overarching statistical appropriateness vs interpretability trade-off, so have added some further discussion in an additional paragraph, line 297:

“The methods evaluated in this manuscript highlight an inherent trade-off in study design between model complexity/appropriateness of assumptions, and interpretability. Typically, simpler methods are more easily communicated, particularly to audiences without statistical training. More complex models may more accurately represent the data but the underlying message can be lost. Ultimately, method selection should strike an appropriate balance for the study objectives.”

Minor Comments

Figure 2 could benefit from scaling the axes to better highlight the initial sharp increase in MCS.

Response: We have reviewed Figure 2 and believe that the scaling of the axes sufficiently represents the MCS and PCS trajectories. The scales chosen are by intentionintentially the same on each y axis (increments of 2.5), but start from a different value. The initial increase in MCS is apparent and more clearly communicated with the numerical annotations.

Ensure consistent terminology for physical health-related quality of life versus PCS throughout the text.

Response: We have now ensured that we exclusively refer to our results as MCS and PCS. As these are representative of metal and physical health-related quality of life, we also refer to these terms in the discussion.

Please clarify in the Methods section whether "time" has been coded in months as numeric or categorical for continuous-time wGEE analyses.

Response: We have added some clarification to the methods to highlight the use of a time variable in numeric months, line 168: “To compare alternative non-linear modelling approaches for PRO analyses, the wGEE analysis was repeated with time as a continuous, numeric, non-linear variable”

Where both AIC and QIC are referenced, ensure the rationale for their use is clearly stated.

Response: We thank the reviewer for picking up on readers’ likely unfamiliarity with the QIC. We assume that readers are likely to have more understanding of commonly used AIC, so we have left the introduction to that as it is: line 146 “we then validated the covariate selection using the Akaike information criterion (AIC) and including only those that improved the model fit (lower AIC)”. However, we have added further explanation to the introduction of the QIC, line 164: “The QIC was developed as a modification of the AIC to apply to models fit by the GEE approach. Therefore, in this manuscript we use the QIC for GEE analyses and AIC for LMM analyses.”

Data Availability

Although the ethical constraints regarding patient data are understandable, the manuscript does not fully meet PLOS ONE data availability requirements. Only the sample R code is publicly available, and the underlying dataset cannot be shared. It may be helpful to provide more detailed guidance to researchers on how to access restricted data (e.g., through a controlled application process), if applicable.

Response: We have consulted with the Gilead Sciences Data Sharing team and have added a company-approved statement on data availability to the end of the manuscript, line 378:

“Gilead Sciences shares anonymised individual patient data upon request or as required by law or regulation with qualified external researchers based on submitted curriculum vitae and reflecting non conflict of interest. The request proposal must also include a statistician. Approval of such requests is at Gilead Science’s discretion and is dependent on the nature of the request, the merit of the research proposed, the availability of the data, and the intended use of the data. Data requests should be sent to datasharing@gilead.com.”

Conclusion

Overall, this is a well-executed and valid applied methodological study. After addressing the above clarifications, the manuscript will be further strengthened and will provide clearer guidance for practitioners working with PRO data in observational settings. I recommend acceptance after minor revisions.

Response: We thank this reviewer for their comments and suggestions. We believe that the manuscript is now improved and agree that it will be a valuable tool for researchers utilising observational PRO data.

References

1. Coens C, Pe M, Dueck AC, Sloan J, Basch E, Calvert M, et al. International standards for the analysis of quality-of-life and patient-reported outcome endpoints in cancer randomised controlled trials: recommendations of the SISAQOL Consortium. Vol. 21, The Lancet Oncology. 2020. p. e83–96.

26. Bell ML, Horton NJ, Dhillon HM, Bray VJ, Vardy J. Using generalized estimating equations and extensions in randomized trials with missing longitudinal patient reported outcome data. Psychooncology. 2018;27(9):2125–31.

39. Hamel JF, Saulnier P, Pe M, Zikos E, Musoro J, Coens C, et al. A systematic review of the quality of statistical methods employed for analysing quality of life data in cancer randomised controlled trials. Vol. 83, European Journal of Cancer. 2017. p. 166–76.

---

## [Decision Letter · Decision Letter 2]

2 Mar 2026

A practical evaluation of statistical methods for the analysis of patient reported outcomes in an observational pharmaceutical study

PONE-D-24-44219R2

Dear Dr. Ramroth,

Dear authors,

I read your paper myself and came to conclusion that you addressed all concerns of the reviewers. The paper is in a publishable form now. I want personally to congratulate you on a solid work and I'm looking forward to seeing further advances from your group in this field.

We’re pleased to inform you that your manuscript has been judged scientifically suitable for publication and will be formally accepted for publication once it meets all outstanding technical requirements.

Kind regards,

Eugene Demidenko, Ph.D.

Academic Editor

PLOS One

Additional Editor Comments (optional):

Reviewers' comments:

Reviewer's Responses to Questions

**Comments to the Author**

Reviewer #3: All comments have been addressed

2. Is the manuscript technically sound, and do the data support the conclusions?

Reviewer #3: Yes

3. Has the statistical analysis been performed appropriately and rigorously?

Reviewer #3: Yes

4. Have the authors made all data underlying the findings in their manuscript fully available?

Reviewer #3: Yes

5. Is the manuscript presented in an intelligible fashion and written in standard English?

Reviewer #3: Yes

Reviewer #3: The revised version demonstrates greater conceptual clarity and transparency. In particular, the authors now explicitly state that the comparison of statitical methods was based on consistency with clinically plausible outcome trajectories rather than on accurate information relative to the known ground truth. This clarification reinforces the interpretability and methodological consistency of the conclusions. The discussion of assumptions about missing data, model interpretability, and trade-offs between analytical complexity and communicability is clear, balanced, and well-articulated.

Although patient data cannot be shared publicly due to confidentiality restrictions, the revised statement on data availability adequately explains these restrictions and provides a clear mechanism for controlled access to the data.

In summary, the manuscript meets the PLOS ONE criteria fore scientific soundness, methodological rigour, and clarity.

**Do you want your identity to be public for this peer review?** For information about this choice, including consent withdrawal, please see our Privacy Policy

Reviewer #3: No

---

## [Editor Report · Acceptance letter]

PONE-D-24-44219R2

PLOS One

Dear Dr. Ramroth,

I'm pleased to inform you that your manuscript has been deemed suitable for publication in PLOS One. Congratulations! Your manuscript is now being handed over to our production team.

Kind regards,

on behalf of

Dr. Eugene Demidenko

Academic Editor

PLOS One